# Treatment Strategies for First-Line PD-L1-Unselected Advanced NSCLC: A Comparative Review of Immunotherapy-Based Regimens by PD-L1 Expression and Clinical Indication

**DOI:** 10.3390/diagnostics15151937

**Published:** 2025-07-31

**Authors:** Blerina Resuli, Diego Kauffmann-Guerrero, Maria Nieves Arredondo Lasso, Jürgen Behr, Amanda Tufman

**Affiliations:** 1Department of Medicine V, Ludwig-Maximilians-Universität München University Hospital, Ludwig-Maximilians-Universität München, 81377 Munich, Germany; diego.kauffmannguerrero@med.uni-muenchen.de (D.K.-G.); maria.arredondolasso@med.uni-muenchen.de (M.N.A.L.); juergen.behr@med.uni-muenchen.de (J.B.); amanda.tufman@med.uni-muenchen.de (A.T.); 2Comprehensive Pneumology Center Munich (CPC-M), German Center for Lung Research (DZL), 81377 Munich, Germany

**Keywords:** non-small cell lung cancer, screening, diagnosis, immune check point inhibitors, PD-L1, meta-analysis

## Abstract

**Background:** Lung cancer remains the leading cause of cancer-related mortality worldwide. Advances in screening, diagnosis, and management have transformed clinical practice, particularly with the integration of immunotherapy and target therapies. **Methods:** A systematic literature search was carried out for the period between October 2016 to September 2024. Phase II and III randomized trials evaluating ICI monotherapy, ICI–chemotherapy combinations, and dual ICI regimens in patients with advanced NSCLC were included. Outcomes of interest included overall survival (OS), progression-free survival (PFS), and treatment-related adverse events (AEs). **Results:** PD-1-targeted therapies demonstrated superior OS compared to PD-L1-based regimens, with cemiplimab monotherapyranking highest for OS benefit (posterior probability: 90%), followed by sintilimab plus platinum-based chemotherapy (PBC) and pemetrexed—PBC. PFS atezolizumab plus bevacizumab and PBC, and camrelizumab plus PBC were the most effective regimens. ICI–chemotherapy combinations achieved higher ORRs but were associated with greater toxicity. The most favorable safety profiles were observed with cemiplimab, nivolumab, and avelumab monotherapy, while atezolizumab plus PBC and sugemalimab plus PBC carried the highest toxicity burdens. **Conclusions:** In PD-L1-unselected advanced NSCLC, PD-1 blockade—particularly cemiplimab monotherapy—and rationally designed ICI–chemotherapy combinations represent the most efficacious treatment strategies. Balancing efficacy with safety remains critical, especially in the absence of predictive biomarkers. These findings support a patient-tailored approach to immunotherapy and highlight the need for further biomarker-driven and real-world investigations to optimize treatment selection.

## 1. Introduction

Lung cancer remains one of the most prevalent malignancies worldwide and continues to be the leading cause of cancer-related mortality in both men and women [1]. Non-small cell lung cancer (NSCLC) accounts for approximately 80–85% of all lung cancer cases, making it the most frequently diagnosed histological subtype [2]. Given the high disease burden and the frequently late-stage presentation, early detection and accurate characterization of pulmonary nodules are critical for improving patient outcomes through timely diagnosis and optimal treatment selection [3]. Although smoking prevalence has declined in many regions, tobacco exposure continues to be the leading cause of lung cancer, accounting for the vast majority of cases—approximately 85% [4]. Nonetheless, a growing proportion of NSCLC diagnoses occur in never-smokers, particularly among women and East Asian populations, pointing to a multifactorial pathogenesis involving genetic predisposition, environmental pollutants, occupational carcinogens, and chronic inflammatory processes [5,6].

Early detection and accurate staging are critical for optimizing treatment in advanced NSCLC, including PD-L1-unselected cases. Advances in imaging, such as low-dose computed tomography (LDCT) and ^18^Fluorodeoxyglucose Positron Emission Tomography (^18^F-FDG PET/CT), have improved early diagnosis and precise staging [7]. Landmark trials like NLST and NELSON support LDCT screening in high-risk populations, though concerns about overdiagnosis and radiation exposure limit broader use [8,9]. These imaging tools also inform staging and treatment planning in advanced disease [10].

In parallel, molecular diagnostics have transformed the clinical landscape of NSCLC, enabling the identification of actionable genetic alterations (e.g., *Epidermal Growth Factor Receptor* (EGFR), Anaplastic Lymphoma Kinase (ALK), *c-ros Oncogene 1* (ROS1), *v-Raf Murine Sarcoma Viral Oncogene Homolog B1* (BRAF), *Mesenchymal–Epithelial Transition Factor* (MET), *Rearranged during Transfection* (RET), and predictive biomarkers such as programmed death ligand-1 (PD-L1) expression and tumor mutational burden (TMB) [11,12,13]. These advances have paved the way for precision oncology, wherein therapeutic decisions are increasingly guided by the molecular and immunological profiles of individual tumors [14].

The advent of immune checkpoint inhibitors (ICIs), particularly those targeting programmed cell death protein 1 (PD-1) and its ligand PD-L1, has revolutionized the management of advanced NSCLC. Agents such as pembrolizumab, nivolumab, atezolizumab, and cemiplimab have demonstrated significant improvements in overall survival, durable response rates, and favorable toxicity profiles compared to conventional chemotherapy [15,16,17]. In patients with high PD-L1 expression (tumor proportion score ≥ 50%), monotherapy with anti–PD-1/PD-L1 agents has become standard first-line therapy [18,19]. However, the majority of real-world patients present with low or indeterminate PD-L1 expression, necessitating the use of combination strategies, including ICIs with platinum-based chemotherapy or dual immune checkpoint blockade (e.g., PD-1 plus Cytotoxic T-Lymphocyte–Associated protein 4 (CTLA-4) inhibition) [20,21].

Despite these therapeutic advances, treatment selection in PD-L1-unselected patients remains challenging, with multiple approved regimens but limited head-to-head comparative data [22]. Additionally, the clinical relevance of emerging biomarkers such as TMB, immune gene signatures, and tumor-infiltrating lymphocytes is still under investigation, and their integration into clinical practice remains limited [23,24,25].

This review aims to provide a comprehensive synthesis of current diagnostic and therapeutic strategies for advanced NSCLC, with a particular focus on ICI-based therapies in patients unselected for PD-L1 expression to evaluate the relative efficacy and safety of ICI monotherapy, chemo-immunotherapy combinations, and dual ICI regimens. Through this integrated approach, we aim to inform evidence-based treatment selection and highlight opportunities for personalized care in the evolving landscape of NSCLC immunotherapy.

## 2. Materials and Methods

### 2.1. Search Strategy and Study Selection

A systematic search of the literature for randomized controlled trials was conducted according to the Preferred Reporting Items for Systematic reviews and Meta-Analysis (PRISMA) guidelines. The literature search on Pubmed-MEDLINE was performed between October 2016 to September 2024. The searching strategy was as follows: ((“Carcinoma, Non-Small-Cell Lung”[Mesh]) OR (((lung[Title/Abstract]) AND (((((cancer*[Title/Abstract]) OR (carcininoma*[Title/Abstract])) OR (neoplasm*[Title/Abstract])) OR (tumour*[Title/Abstract])) OR (tumor*[Title/Abstract]))) AND ((non-small[Title/Abstract]) OR (non-small[Title/Abstract])))) OR (nsclc[Title/Abstract]) OR Immune Checkpoint Inhibitors”[Mesh]) OR (immune checkpoint inhibitor [Title/Abstract])) OR (immune checkpoint inhibitors[ Title/Abstract])) OR (“Programmed Cell Death 1 Receptor”[Mesh])) OR (Programmed Cell Death 1 Receptor[Title/Abstract])) OR (Programmed Cell Death 1 Protein[Title/Abstract])) OR (PD-1 Receptor[Title/Abstract])) OR (“Nivolumab”[Mesh])) OR (Nivolumab[Title/Abstract])) OR (opdivo[Title/Abstract])) OR (BMS-936558 [Title/Abstract])) OR (BMS936558 [Title/Abstract])) OR (BMS 936558[Title/Abstract])) OR (ONO4538[Title/Abstract])) OR (ONO 4538[Title/Abstract])) OR (MDX-1106[Title/Abstract])) OR (MDX1106[Title/Abstract])) OR (MDX 1106[Title/Abstract])) OR (“toripalimab” [Supplementary Concept])) OR (Toripalimab[Title/Abstract])) OR (“pembrolizumab” [Supplementary Concept])) OR (pembrolizumab[Title/Abstract])) OR (Keytruda[Title/Abstract])) OR (lambrolizumab[Title/Abstract])) OR (MK-3475[Title/Abstract])) OR (MK3475[Title/Abstract])) OR (MK 3475[Title/Abstract])) OR (SCH-900475[Title/Abstract])) OR (SCH900475[Title/Abstract])) OR (SCH 900475[Title/Abstract])) OR (“atezolizumab” [Supplementary Concept])) OR (Atezolizumab[Title/Abstract])) OR (MPDL3280A[Title/Abstract])) OR (MPDL-3280A[Title/Abstract])) OR (MPDL 3280A[Title/Abstract])) OR (RG7446[Title/Abstract])) OR (RG-7446[Title/Abstract])) OR (RG 7446[Title/Abstract])) OR (Tecentriq[Title/Abstract])) OR (“ramucirumab” [Supplementary Concept])) OR (Ramucirumab[Title/Abstract])) OR (IMC1121B[Title/Abstract])) OR (IMC-1121B[Title/Abstract])) OR (IMC 1121B[Title/Abstract])) OR (Cyramza[Title/Abstract])) OR (“sintilimab” [Supplementary Concept])) OR (Sintilimab[Title/Abstract])) OR (“camrelizumab” [Supplementary Concept])) OR (camrelizumab[Title/Abstract])) OR (“tislelizumab” [Supplementary Concept])) OR (tislelizumab[Title/Abstract])) OR (durvalumab [Supplementary Concept])) OR (durvalumab[Title/Abstract])) OR (“CTLA-4 Antigen”[Mesh])) OR (CTLA-4 Antigen[Title/Abstract])) OR (CTLA 4 Antigen[Title/Abstract])) OR (CTLA-4[Title/Abstract])) OR (CD152 Antigens[Title/Abstract])) OR (Cytotoxic T-Lymphocyte-Associated Antigen 4[Title/Abstract])) OR (Cytotoxic T Lymphocyte-Associated Antigen 4[Title/Abstract])) OR (“Ipilimumab”[Mesh])) OR (Ipilimumab[Title/Abstract])) OR (yervoy[Title/Abstract])) OR (“Bevacizumab”[Mesh])) OR (Bevacizumab*[Title/Abstract])) OR (Mvasi[Title/Abstract])) OR (Avastin[Title/Abstract]). Keywords “non-small cell lung cancer, immune check point inhibitors, PD-1, PD-L1” were used for searching in Cochrane Central Register of Controlled Trials and SCOPUS. No poster presentation or abstract was included. The systematic review followed the recommendations of the Preferred Reporting Items for Systematic Reviews and Meta-Analysis (PRISMA). The protocol was not registered. Figure 1 summarizes the study selection process.

### 2.2. Study Selection Criteria and Data Extraction

Phase II and III randomized controlled trials in adults (men and women) with advanced NSCLC evaluated for overall survival (OS), progression-free survival (PFS), and adverse events (AEs) according to Common Terminology Criteria for Adverse Events (CTCAE 4.0); and receiving ICI monotherapy or these in combination were included in our analysis.

Data were extracted independently by authors and entered in a standardized, predesigned Microsoft Excel form. The following data were recorded: author, publication year, study design, number of total patients, median age of patients, stage, histology, kind of treatment (ICI monotherapy or ICI combination), Hazard Ratio (HR) of OS, and PFS. Each author assessed the quality of reporting.

### 2.3. Statistical Analysis

A random-effect meta-analysis model was applied, and the inverse-variance weighting was used to pool estimates of the included studies. The meta-analysis was conducted by Rev-Man 5.4 software. The importance of the observed value of I^2^ depends on the magnitude and direction of effects and the strength of evidence for heterogeneity. Statistical significance was defined at the 0.05 level. Time-to-event outcomes (OS and PFS) were analyzed using the generic inverse variance method and expressed as log HRs with corresponding 95% confidence intervals (CIs). Dichotomous outcomes, including adverse events, were analyzed using inverse variance weighting to estimate risk ratios (RRs) with 95% CIs.

### 2.4. Ethical Considerations

This study involved secondary analysis of publicly available, previously published trial data and therefore did not require ethical approval or patient consent.

## 3. Results

### 3.1. Literature Search and Included Studies

Overall, 4632 records were identified using the search strategy. After screening the titles and abstracts, 4150 records were excluded based on the type of article (review, case report, clinical trials) or irrelevant content. Of the remaining 125 potentially relevant studies, we excluded 80 papers for not having enough data available or for not meeting our inclusion criteria. Finally, we selected 22 randomized controlled clinical trial articles published worldwide between 2016 and 2024. The selection process is shown in Figure 1 and all the studies included are shown in Table 1 [15,17,19,26,27,28,29,30,31,32,33,34,35,36,37,38,39,40,41,42,43,44].

### 3.2. Overall Survival (OS)

Twenty-two first-line immunotherapy-based regimens derived from randomized controlled trials (RCTs) were selected for comparative evaluation based on clinical relevance and data completeness. Considerable heterogeneity in overall survival outcomes was observed across these regimens, driven by differences in treatment strategy (monotherapy vs. combination therapy), histologic subtype (squamous vs. non-squamous), and PD-L1 expression levels [15,17,19,26,27,28,29,30,31,32,33,34,35,36,37,38,39,40,41,42,43,44]. Stratifying results by PD-L1 expression enables a more precise understanding of treatment efficacy across clinically distinct subpopulations.

#### 3.2.1. PD-L1 ≥ 50%

In patients with high PD-L1 expression (≥50%), ICI monotherapy provided the most pronounced survival benefit relative to platinum-doublet chemotherapy, with multiple trials demonstrating superiority in this biomarker-enriched subgroup.

In the EMPOWER-Lung 01 trial, cemiplimab monotherapy achieved the most favorable OS outcome, with a HR of 0.59 (95% CI, 0.48–0.072) [17]. At a median follow-up of 59.6 months, median OS was 26.1 months in the cemiplimab arm versus 13.3 months in the chemotherapy arm. 

Although atezolizumab monotherapy in IMpower110 initially showed OS benefit in high PD-L1 expressors (HR, 0.59; 95% CI, 0.40–0.89), this advantage diminished with longer follow-up. At final analysis, OS was not statistically significant in the high-or-intermediate PD-L1 wild-type population (HR, 0.87; 95% CI, 0.66–1.14), and formal testing in the overall PD-L1 population was not conducted (HR, 0.85; 95% CI, 0.69–1.04) [29].

In contrast, dual-ICI regimens without chemotherapy failed to improve survival in PD-L1 ≥ 50% patients. In KEYNOTE-598, adding ipilimumab to pembrolizumab in PD-L1 ≥ 50% NSCLC did not improve OS (21.4 vs. 21.9 months; HR, 1.08; 95% CI, 0.85–1.37; *p* = 0.74), and increased toxicity led to early trial termination for futility [35].

In PD-L1 ≥ 50%, ICI plus chemotherapy combinations have also demonstrated efficacy. In EMPOWER-Lung 03, cemiplimab plus chemotherapy yielded an OS HR of 0.65 versus chemotherapy alone [37,38,44]. Likewise, in KEYNOTE-189 and KEYNOTE-407, pembrolizumab plus platinum-based chemotherapy achieved OS HRs of 0.68 (non-squamous NSCLC) and 0.64 (squamous NSCLC), respectively [18,33,34]. These findings support the use of chemo-immunotherapy in select high–PD-L1 patients, particularly those with aggressive disease features or high tumor burden requiring rapid disease control.

#### 3.2.2. PD-L1 < 50%

In patients with low or negative PD-L1 expression (<50%), ICI monotherapy has demonstrated limited OS benefit. In this subgroup, outcomes with single-agent immunotherapy were generally inferior or non-superior to chemotherapy, and thus combination regimens are preferred.

ICI plus chemotherapy regimens have consistently shown robust and reproducible OS improvements in PD-L1 < 50% populations. Among the most effective was pembrolizumab plus chemotherapy, which achieved OS HRs of 0.65 (PD-L1 1–49%) and HR 0.55 (PD-L1 < 1) (KEYNOTE-189) and 0.61 in squamous histologies (KEYNOTE-407 PD-L1 1–49%) and 0.83 PD-L1 < 1% [18,33,34]. These benefits were observed across PD-L1 strata, including PD-L1 1–49% and <1% subgroups.

Emerging PD-1 inhibitors, such as camrelizumab and tislelizumab, have also demonstrated strong OS efficacy when combined with chemotherapy. Camrelizumab plus platinum-based chemotherapy reported HRs ranging from 0.55 to 0.72 across histologies [37,38], while tislelizumab plus chemotherapy in LU2024 achieved an OS HR of 0.68 [39,44]. Cemiplimab plus chemotherapy in EMPOWER-Lung 03 (HR, 0.65) further supports the role of PD-1 blockade with histology-tailored chemotherapy in this subgroup [42].

Dual-ICI plus chemotherapy regimens showed intermediate but durable OS benefit. In Figure, nivolumab plus ipilimumab with limited chemotherapy achieved a 5-year OS HR of 0.73 (95% CI, 0.62–0.85) versus chemotherapy [21,28], with consistent benefit across subgroups. Five-year OS rates were 22% vs. 8% in PD-L1 < 1%, 18% vs. 11% in PD-L1 ≥ 1%, 18% vs. 7% in squamous, and 19% vs. 12% in non-squamous NSCLC.

In POSEIDON, durvalumab plus tremelimumab and chemotherapy yielded a 5-year OS of 15.7% vs. 6.8% (HR, 0.76; 95% CI, 0.64–0.89), including in PD-L1-negative and *Serine/Threonine Kinase 11* (STK11)-, *Kelch-like ECH-associated Protein 1* (KEAP1)-, or *Kirsten Rat Sarcoma Viral Oncogene Homolog* (KRAS)-mutant tumors [13,20,27,41]. Durvalumab plus chemotherapy alone provided a more modest benefit (HR, 0.84; 5-year OS, 13.0%).

Finally, atezolizumab plus chemotherapy combinations demonstrated modest and variable OS benefit across histologies. In IMpower130, atezolizumab plus carboplatin and nab-paclitaxel significantly improved OS versus chemotherapy (HR, 0.79; 95% CI, 0.64–0.98) in non-squamous NSCLC [19]. In IMpower132, the OS benefit was not statistically significant (HR, 0.81; 95% CI, 0.64–1.05) [30]. In IMpower131, conducted in squamous NSCLC, no OS improvement was observed (HR, 0.92; 95% CI, 0.76–1.12) [31]. These findings suggest that atezolizumab-based regimens may serve as alternatives to pembrolizumab-based combinations, particularly in non-squamous, PD-L1-low or unselected populations

The data of the studies underscore the critical role of PD-L1 expression in guiding first-line treatment selection for advanced NSCLC. In PD-L1 ≥ 50%, ICI monotherapy—particularly with cemiplimab or pembrolizumab—offers the most favorable benefit–risk profile, with long-term survival benefits and lower toxicity. However, chemo-immunotherapy combinations remain effective and may be preferred in patients with high disease burden or poor prognostic features.

In PD-L1 < 50%, ICI plus chemotherapy combinations represent the current standard of care, offering the most consistent and clinically meaningful OS improvements. Among these, pembrolizumab, camrelizumab, and tislelizumab-based regimens appear particularly efficacious. Dual-ICI plus chemotherapy regimens, such as those investigated in CheckMate 9LA and POSEIDON, offer viable alternatives in broader patient populations, including those with immunologically resistant tumors. Figure 2 summarizes the overall survival outcomes.

### 3.3. Progression-Free Survival

Across 22 randomized clinical trials evaluating first-line immunotherapy-based regimens in advanced NSCLC, PFS outcomes demonstrated that ICI plus chemotherapy combinations consistently conferred the most substantial and reproducible benefits in delaying disease progression. These findings confirm that chemotherapy is essential for optimal disease control in PD-L1 low/negative populations, with substantial additive benefit from immunotherapy. The magnitude and consistency of PFS benefit, however, varied significantly according to PD-L1 expression status, underscoring the biomarker’s role in shaping treatment strategy. Figure 3 summarizes the progression-free survival outcomes.

#### 3.3.1. PD-L1 ≥ 50%

In patients with PD-L1 ≥ 50%, ICI monotherapy provided significant PFS benefit over chemotherapy. Pembrolizumab monotherapy in KEYNOTE-024 showed a PFS HR of 0.50 (95% CI, 0.37–0.68) [15], cemiplimab in EMPOWER-Lung 01 achieved 0.50 (95% CI, 0.41–0.61) [17], and atezolizumab in IMpower110 reported a HR of 0.63 (95% CI, 0.45–0.88) [29].

ICI plus chemotherapy regimens offered further PFS enhancement in this subgroup. In KEYNOTE-189, pembrolizumab plus pemetrexed-platinum showed a HR of 0.35 [33]; KEYNOTE-407 demonstrated HRs of 0.48 (PD-L1 ≥ 50%), 0.60 (PD-L1 1–49%), and 0.70 (PD-L1 < 1%) [34]. EMPOWER-Lung 03 reported a HR of 0.48 in PD-L1 ≥ 50%, while LU2024 (RATIONALE-304) showed a HR of 0.29 for tislelizumab in the same subgroup [39,42]. These data underscore that even among high expressors, adding chemotherapy improves early disease control.

#### 3.3.2. PD-L1 < 50%

In PD-L1 < 50% populations, ICI monotherapy demonstrated minimal PFS benefit. CheckMate 598 (HR, 1.06) [35] and MYSTIC (HR, 1.05) [36] confirmed the limited efficacy of monotherapy in this setting.

ICI plus chemotherapy regimens consistently yielded substantial PFS improvements. In PD-L1 1–49%, PFS HRs were 0.60 (KEYNOTE-407), 0.57 (KEYNOTE-189), 0.48 (EMPOWER-Lung 03), and 0.67 (IMpower110) [33,34,42]. In PD-L1 < 1%, corresponding HRs were 0.70 (KEYNOTE-407), 0.67 (KEYNOTE-189), 0.73 (EMPOWER-Lung 03), and 0.84 (IMpower110) [33,34,42]. In the RTIONALE 304 study, the HR was 0.90 (PD-L1 1–49%) and 0.83 (PD-L1 < 1%) [39]. Sugemalimab (Zhou et al., 2022) reported a PFS HR of 0.49 [40]; camrelizumab in CameL-sq achieved 0.37 [38]; and atezolizumab-based combinations in IMpower130, 132, and 150 showed HRs of 0.64, 0.60, and 0.62, respectively [19,31,32].

#### 3.3.3. PD-L1 Unselected and Molecular Subgroups

Dual-ICI plus chemotherapy regimens demonstrated intermediate PFS benefit across unselected populations. CheckMate 9LA reported HRs of 0.70; 95% CI, 0.60–0.83, regardless of PD-L1 or histology [28]. In POSEIDON, durvalumab plus tremelimumab and chemotherapy achieved an overall PFS HR of 0.72 (95% CI, 0.60–0.86), with consistent benefit across PD-L1 levels and improved outcomes in non-squamous histology [41].

These data suggest that dual-ICI plus chemotherapy offers a viable strategy in PD-L1 unselected or biomarker-negative patients, particularly those with poor-risk molecular profiles (e.g., STK11, KEAP1, KRAS mutations), where monotherapy may be inadequate.

### 3.4. Adverse Events

Across the evaluated trials, toxicity profiles varied substantially by treatment strategy. ICI monotherapy regimens, including pembrolizumab, atezolizumab, and cemiplimab, were consistently associated with the lowest incidence of grade ≥ 3 treatment-related adverse events (TRAEs), typically ranging between 15% and 30%. In KEYNOTE-024, pembrolizumab monotherapy resulted in grade ≥ 3 TRAEs in 26.6% of patients, compared to 53.3% with chemotherapy [27]. Similarly, cemiplimab monotherapy in EMPOWER-Lung 01 had a grade ≥ 3 TRAE rate of 28% [17], while atezolizumab monotherapy in IMpower110 reported 30.1% [29], confirming the favorable safety profile of PD-(L)1 inhibitors in isolation.

By contrast, ICI plus chemotherapy combinations demonstrated higher toxicity burdens, with grade ≥ 3 TRAE rates generally ranging from 50 to 73%, largely driven by the chemotherapy backbone. In KEYNOTE-189 and KEYNOTE-407, pembrolizumab plus platinum-doublet chemotherapy led to grade ≥ 3 TRAEs in 67.2% and 69.8% of patients, respectively [18,34]. Similarly, in IMpower130 and IMpower150, atezolizumab-based combinations reported grade ≥ 3 toxicity rates of 73% and 64%, respectively [19,32]. Chinese trials of novel PD-1 inhibitors showed comparable toxicity profiles (Figure 4): camrelizumab plus chemotherapy in CameL-Sq resulted in grade ≥ 3 TRAEs in 61.2% of patients [38], sugemalimab plus chemotherapy in GEMSTONE-302 reported 61.3% [40], tislelizumab plus chemotherapy in RATIONALE 307 and RATIONALE 304 showed grade ≥ 3 TRAEs of 63.5% and 61.7%, respectively [39,44], and camrelizumab plus chemotherapy in CameL-Nsq reported 61.3% [37].

Dual-ICI combinations, particularly when used without chemotherapy, were associated with increased immune-related toxicity. In CheckMate 227, nivolumab plus ipilimumab led to grade ≥ 3 TRAEs in 32.8% of patients—higher than monotherapy but lower than chemo-ICI combinations [20]. However, when combined with chemotherapy, as in CheckMate 9LA, the overall grade ≥ 3 toxicity rate rose to 47%, and treatment-related discontinuation occurred in 19% of patients [28]. Similarly, in the POSEIDON trial, durvalumab plus tremelimumab with chemotherapy resulted in grade ≥ 3 TRAEs in 62.8% of patients, compared to 48.4% in the chemotherapy-only arm [41].

Notably, treatment discontinuation due to TRAEs was most frequent in dual-ICI arms. For instance, in CheckMate 598 (ipilimumab + pembrolizumab), 35% of patients discontinued due to toxicity [35]. In contrast, discontinuation rates were much lower for monotherapy arms, typically below 10% [15,17,27].

Overall, these findings underscore the superior safety profile of ICI monotherapy, the predictable but significant toxicity burden of ICI–chemotherapy combinations, and the heightened risk of immune-mediated and cumulative toxicity with dual-ICI strategies, particularly when used in combination with chemotherapy.

## 4. Discussion

This comprehensive comparative analysis of first-line ICI-based therapies for advanced NSCLC in PD-L1-unselected populations reaffirms the paradigm shift toward immunotherapy-based regimens in international guidelines. The 2024 National Comprehensive Cancer Network (NCCN), American Society of Clinical Oncology (ASCO), and European Society of Clinical Oncology (ESMO) guidelines endorse ICI plus chemotherapy as the preferred first-line option in patients with PD-L1 < 50%, with pembrolizumab-based combinations established as the standard of care across both squamous and non-squamous histologies. These recommendations are supported by our synthesis, which demonstrates that ICI–chemotherapy combinations consistently provide the most robust clinical benefit across OS and PFS endpoints in the absence of high PD-L1 expression [17,18,26,29,31,32,37,38,39,40,43,44].

Among combination strategies, pembrolizumab plus chemotherapy (KEYNOTE-189 and KEYNOTE-407) remains the benchmark, demonstrating consistent survival gains across histologic subtypes. Importantly, regimens incorporating novel PD-1 inhibitors developed in China—including camrelizumab, tislelizumab, and sugemalimab—have shown comparable efficacy and are expanding global access, particularly in squamous NSCLC, where camrelizumab plus chemotherapy achieved a PFS HR as low as 0.37 [38,39,40,44].

ICI monotherapy, although well tolerated, is effective primarily in patients with PD-L1 ≥ 50% tumors. Cemiplimab, atezolizumab, and pembrolizumab monotherapies have demonstrated significant benefits in this subgroup [15,17,29]. However, their limited efficacy in PD-L1-low or heterogeneous tumors highlights the inadequacy of PD-L1 as a standalone biomarker. Similarly, dual-ICI regimens such as ipilimumab plus pembrolizumab (CheckMate 598) and durvalumab ± tremelimumab (MYSTIC) failed to improve outcomes over chemotherapy, reflecting the challenges of non-stratified immunotherapy approaches in frontline settings [35,36].

Dual-ICI combinations with limited chemotherapy, such as nivolumab plus ipilimumab in CheckMate 9LA, represent a potential compromise between efficacy and toxicity. However, the magnitude of benefit remains inferior to pembrolizumab-based chemo-immunotherapy [20,28]. Grade ≥ 3 TRAEs occurred in nearly half of patients, with discontinuation rates exceeding 15%. Similarly, the POSEIDON regimen (durvalumab plus tremelimumab with chemotherapy) yielded intermediate efficacy but notable immune-related toxicity, suggesting these regimens may be most suitable for patients with aggressive disease and preserved performance status [41].

From a safety perspective, ICI monotherapy remains the most favorable, with grade ≥ 3 TRAEs typically <30% and discontinuation rates < 10% [15,17,29]. By contrast, chemo-ICI combinations incur higher toxicity (grade ≥ 3 TRAEs in 60–70%), largely attributable to chemotherapy, underscoring the importance of careful patient selection and supportive care [18,19,33,35]. Dual-ICI regimens, particularly in the absence of chemotherapy, are associated with increased immune-related adverse events, limiting their role in routine practice outside of investigational or biomarker-enriched settings [28,31,41].

Despite the transformative impact of ICIs in advanced NSCLC, significant challenges impede their optimal use in real-world clinical practice. First, pivotal trials disproportionately enrolled highly selected patients, excluding those with ECOG performance status ≥ 2, active or untreated brain metastases, autoimmune disorders, or significant comorbidities—subgroups that represent a substantial portion of the NSCLC population encountered in routine oncology clinics. Consequently, the generalizability of trial findings remains limited.

Second, immune-related adverse events (irAEs), while manageable in well-resourced trial settings, present unique challenges in daily practice, particularly in centers without multidisciplinary expertise in immunotherapy toxicity management. IrAEs can also lead to treatment discontinuation or long-term morbidity, complicating care in older or frail patients.

Third, disparities in access to novel ICIs and supportive care infrastructure amplify global inequities, especially in low- and middle-income countries where pembrolizumab and other checkpoint inhibitors are not universally available or affordable. Even in high-income settings, drug costs and reimbursement policies can constrain patient access.

Finally, reliance on PD-L1 as the primary biomarker fails to account for tumor and host heterogeneity. PD-L1 expression is often dynamic, spatially heterogeneous, and inconsistently predictive of response, particularly in tumors with intermediate PD-L1 levels (1–49%). Alternative biomarkers such TMB have demonstrated limited utility in prospective studies and are no longer broadly recommended in guidelines [14,23]. Emerging approaches such as spatial transcriptomics, multiplex immunohistochemistry (IHC), single-cell RNA sequencing, and digital pathology are beginning to offer insights into the tumor-immune microenvironment that may better predict immunotherapy response. Liquid biopsy-based assays, including circulating tumor DNA (ctDNA) and immune profiling, may allow for real-time monitoring of treatment response and resistance. Ongoing trials such as Lung-MAP, TRACERx, KEYNOTE-495 (KEYLYNK-005), and NeoIMmuno are evaluating the integration of novel biomarkers (e.g., TMB-high, immune cell signatures, ctDNA clearance, T-cell clonality) into treatment algorithms [45,46,47]. These studies may pave the way for biomarker-driven treatment personalization in the coming years.

To overcome these barriers and optimize immunotherapy for NSCLC, a shift toward precision-guided strategies is essential. Integrating dynamic and spatially resolved biomarkers such as multiplex immunohistochemistry, spatial transcriptomics, and single-cell RNA sequencing may enhance understanding of the tumor-immune microenvironment and better predict treatment responses. Liquid biopsy approaches—including ctDNA dynamics and immune profiling—could enable real-time monitoring of response and resistance. Targeting alternative checkpoints (e.g., lymphocyte Activation Gene-3 (LAG-3), T cell immunoreceptor with Ig and ITIM domains (TIGIT), or immunosuppressive pathways (e.g., Vascular Endothelial Growth Factor (VEGF), adenosine axis) may help overcome resistance and broaden the population benefiting from ICIs. Combining ICIs with agents that modulate the tumor microenvironment offers a promising strategy for converting “cold” tumors into “hot” ones. Concepts such as induction chemoimmunotherapy followed by ICI maintenance, limited-duration ICI therapy in responders, and sequential use of chemotherapy and immunotherapy warrant investigation to balance efficacy, toxicity, and cost.

Pragmatic trials focusing on underrepresented populations—elderly patients, those with poor performance status, autoimmune disorders, or brain metastases—are critical to expanding the applicability of immunotherapy in real-world settings. The emergence of PD-1 inhibitors developed in Asia (e.g., camrelizumab, tislelizumab) and biosimilars could democratize access to immunotherapy, provided their comparative efficacy and safety are rigorously established.

## 5. Conclusions

In summary, chemoimmunotherapy remains the cornerstone of first-line treatment for advanced NSCLC unselected for PD-L1 expression. Stratified analyses clearly demonstrate that the magnitude of benefit in both OS and PFS is greatest in PD-L1 < 50% populations when chemotherapy is combined with PD-1/PD-L1 blockade, whereas ICI monotherapy remains optimal primarily in PD-L1 ≥ 50% patients with favorable disease features.

However, bridging the gap between clinical trial efficacy and real-world practice requires innovation in biomarker development, treatment personalization, and global access equity. The next frontier in immuno-oncology lies in dynamic, biomarker-informed strategies tailored to individual tumor–immune interactions, enabling a shift from empirical combination regimens to precision-guided therapies that optimize outcomes while minimizing toxicity. Emerging technologies and adaptive trial designs will be essential to realize the full potential of immunotherapy and move beyond one-size-fits-all paradigms in NSCLC.

## Figures and Tables

**Figure 1 diagnostics-15-01937-f001:**
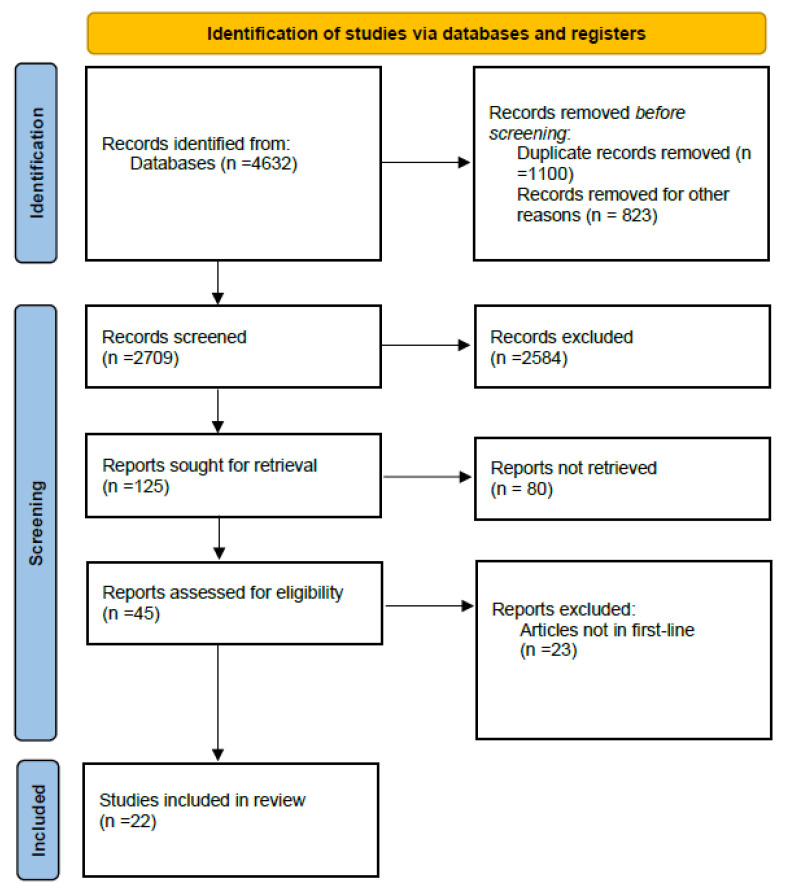
Flowchart of the selection process.

**Figure 2 diagnostics-15-01937-f002:**
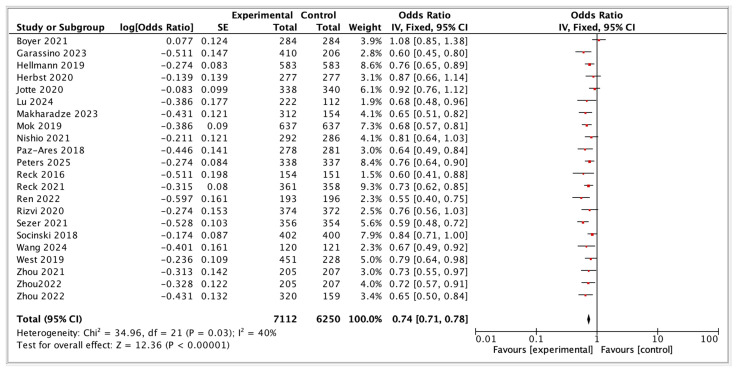
Overall survival of ICI-based Regimens [15,17,19,26,27,28,29,30,31,32,33,34,35,36,37,38,39,40,41,42,43,44].

**Figure 3 diagnostics-15-01937-f003:**
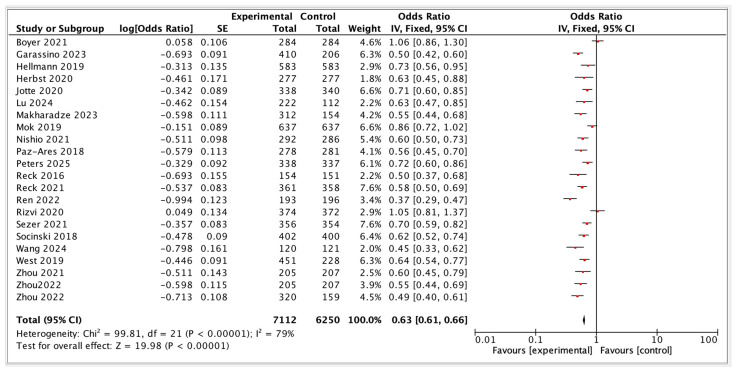
Progression-free survival of ICI-based Regimens [15,17,19,26,27,28,29,30,31,32,33,34,35,36,37,38,39,40,41,42,43,44].

**Figure 4 diagnostics-15-01937-f004:**
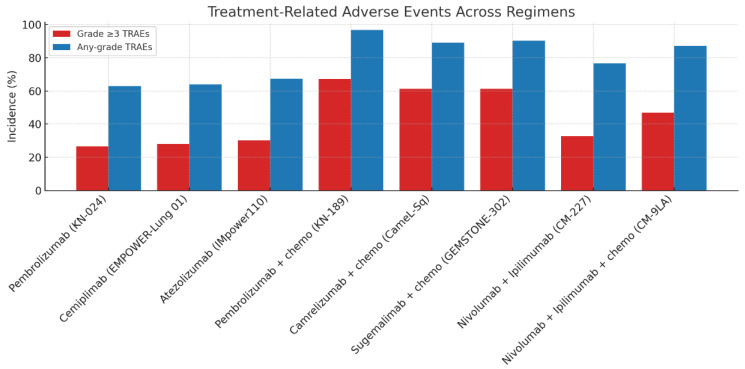
Toxicity profiles of ICI-base Regimens.

**Table 1 diagnostics-15-01937-t001:** Studies analyzed in our review.

Trial (Phase, Design)	Sample Size (Median Age [Years])	Stage (Male/Female)	HR OS	HR PFS	Histology
Mok et al., 2019 [26]	637/637 (63.0/63.0)	NR (902/372)	0.68 [95% CI: 0.57–0.81]	0.86 [95% CI: 0.72–1.02]	NSCLC
West et al., 2019 [19]	451/228 (64/65)	IV (400/279)	0.79 [95% CI 0.64–0.98]	0.64 [95% CI 0.54–0.77]	Non-squamous
Hellmann et al., 2019 [27]	583/583 (64/64)	IV (778/388)	0.76 [95% CI: 0.65–0.90]	0.73 [95% CI, 0.56 to 0.95]	NSCLC
Reck et al., 2021 [28]	361/358 (65/65)	IV (215/504)	0.73 [95% CI 0.62–0.85]	0.70 [95% CI 0.60–0.83]	NSCLC
Sezer et al., 2021 [17]	356/354 (63/64)	IV (606/104)	0.59 [0.48–0.72]	0.50 [0.41–0.61]	NSCLC
Herbst et al., 2020 [29]	277/277 (64/65)	IV (389/165)	0.87 [95% CI, 0.66 to 1.14]	0.63 [95% CI, 0.45 to 0.88]	NSCLC
Jotte et al., 2020 [30]	338/343/340 (66/65/65)	IV (835/186)	0.92 [95% CI: 0.76–1.12]	0.71 [95% CI: 0.60–0.85]	Squamous
Nishio et al., 2021 [31]	292/286 (64/63)	IV (384/194)	0.81 [95% CI: 0.64–1.03]	0.60 [95% CI: 0.49–0.72]	Non-squamous
Socinski et al., 2018 [32]	402/400/400 (63/63/63)	IV (720/482)	0.84 [95% CI, 0.71–1.01]	0.62 [95% CI, 0.52–0.74]	Non-squamous
Reck et al., 2016 [15]	154/151 (64.5/66.0)	IV (187/118)	0.60 [95% CI, 0.41 to 0.89]	0.50 [95% CI, 0.37 to 0.68]	NSCLC
Garassino et al., 2023 [33]	410/206 (65/63.5)	IV (363/253)	0.60 [95% CI, 0.50– 0.72]	0.50 [95% CI, 0.42 to 0.60]	Non-squamous
Paz-Ares et al., 2018 [34]	278/281 (65/65)	IV (455/104)	0.64 [95% CI, 0.49 to 0.85]	0.56 [95% CI, 0.45 to 0.70]	Squamous
Boyer et al., 2021 [35]	284/284 (64/65)	IV (393/375)	1.08 [95% CI, 0.85 to 1.37]	1.06 [95% CI, 0.86 to 1.30]	NSCLC
Rizvi et al., 2020 [36]	374/372/372 (64/65/64.5)	IV (772/346)	0.76 [97.54% CI, 0.56–1.02]	1.05 [99.5% CI, 0.72–1.53]	NSCLC
Zhou et al., 2022 [37]	205/207 (59/61)	IIIB-IV (295/117)	0.72 [95% CI: 0.57–0.92]	0.55 [95% CI: 0.44–0.69]	Non-squamous
Ren et al., 2022 [38]	193/196 (64/62)	IIIB-IV (359/30)	0.55 [95% CI 0.40–0.75]	0.37 [95% CI 0.29–0.47]	Squamous NSCLC
Wang et al., 2024 [39]	120/119/121 (60/63/62)	IIIB–IV (330/30)	0.67 [95% CI 0.49–0.92]	0.45 [95% CI 0.33–0.62]	Squamous NSCLC
Zhou et al., 2022 [40]	320/159 (62/64)	IV (383/96)	0.65 [95%CI, 0.50–0.84]	0.49 [0.40–0.61]	NSCLC
Peters et al., 2025 [41]	338/338/337 (63/64.5/64)	IV (770/243)	0.76 [95% CI, 0.64–0.89]	0.72 [95% CI, 0.60 to 0.86]	NSCLC
Makharadze et al., 2023 [42]	312/154 (63/63)	IIIB-IV (400/66)	0.65 [95% CI 0.51–0.82]	0.55 [95%CI 0.44–0.68]	NSCLC
Zhou et al., 2021 [43]	205/207 (64/62)	IIIB/IV (276/136)	0.73 [95% CI: 0.55–0.96]	0.60 [0.45–0.79]	Non-squamous NSCLC
Lu et al., 2024 [44]	222/112 (62/62)	IIB/IV (247/87)	0.68 [95% CI 0.48–0.96]	0.63 [95% CI 0.47–0.86]	Non-squamous NSCLC

## Data Availability

The data presented in this study are available in this article.

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
