# Peer review of "Treatment Strategies for First-Line PD-L1-Unselected Advanced NSCLC: A Comparative Review of Immunotherapy-Based Regimens by PD-L1 Expression and Clinical Indication"

_diagnostics, 2025, doi:10.3390/diagnostics15151937_

Round 1
Reviewer 1 Report
Comments and Suggestions for Authors
The authors have evaluated very interesting topic about treatment options in PD-L1 unselected advanced NSCLC.
First of all, the article is written on cancers template and it is not prepared by the journal’s instructions. Please correct that. Further the references are not marked appropriately, lines 156, 157, 172, 242. Please check in whole manuscript.
In the introduction, you stated that the smoking is responsible for 85% of cases, you should write it in other way.
In the majority of included trials, there are data about PD-L1 expression and also conclusions accordingly. I suggest to the authors to divide studies according to the PD-L1 expression and their indications. All studies are conducted in the first line setting but not for the same indications, thus it should be separated accordingly.
According to the above mentioned please reconsider to change the title.
The results are unclear, it would be much clearer to present them according to the studies indications and PD-L1 expression.
Please rewrite the conclusion according to the suggestions for the results.
Figures and tables are at the end of the manuscript and that is not according to the journal’s instructions. For all figures and table- it should be better described in the headings what the table/figure represents. In a table the column line of therapy is not necessary since all analyzed studies are in first line setting. In the column treatment the doses of the drugs are also unnecessary. In my opinion the table should be divided in two, one should contain the results for mono- immunotherapy and other for combination. Also it should be marked in the text the place of figures and tables.
Comments on the Quality of English LanguageIn some parts of the manuscripts the sentences are unclear, for example line 175- select monotherapy...
Author Response
-
Reviewer 1
- The authors have evaluated very interesting topic about treatment options in PD-L1 unselected advanced NSCLC.
- Author response:: We sincerely thank the reviewer for recognizing the relevance and interest of our work. The therapeutic landscape for advanced NSCLC in PD-L1 unselected populations remains complex and evolving, and we aimed to provide a comprehensive and balanced evaluation of current immunotherapy-based options. We appreciate your positive feedback and hope that our analysis contributes meaningfully to the ongoing discussion in this area.
- First of all, the article is written on cancers template and it is not prepared by the journal’s instructions. Please correct that. Further the references are not marked appropriately, lines 156, 157, 172, 242. Please check-in whole manuscript.
-Author response: We thank the reviewer for pointing this out. We apologize for the oversight in using the Cancers template. We have now reformatted the manuscript according to the specific guidelines of the target journal. In addition, we have carefully reviewed the entire manuscript and corrected all improperly marked references, including those at lines 156, 157, 172, and 242, to ensure consistency and accuracy in citation formatting throughout the text.
- In the introduction, you stated that the smoking is responsible for 85% of cases, you should write it in other way.
- Author response: We thank the reviewer for this suggestion. To improve clarity and accuracy, we have revised the sentence in the Introduction as follows:
“Although smoking prevalence has declined in many regions, tobacco exposure continues to be the leading cause of lung cancer, accounting for the vast majority of cases—approximately 85%.”- In the majority of included trials, there are data about PD-L1 expression and also conclusions accordingly. I suggest to the authors to divide studies according to the PD-L1 expression and their indications. All studies are conducted in the first line setting but not for the same indications, thus it should be separated accordingly.
- Author response: We thank the reviewer for this valuable suggestion. We agree that stratifying studies based on PD-L1 expression and their specific treatment indications enhances clarity and interpretability. Accordingly, we have revised the Results section to categorize the included trials by PD-L1 subgroups (e.g., <1%, 1–49%, ≥50%) where available, and by their intended clinical indications (e.g., monotherapy, chemo-ICI combinations, dual-ICI regimens). These modifications are also discussed in the revised Discussion section to better contextualize efficacy outcomes across different PD-L1 expression levels and treatment strategies.
- According to the above mentioned please reconsider to change the title.
- Author response: We thank the reviewer for the helpful suggestion. In response, we have revised the title to more accurately reflect the structure of the review and the inclusion of stratified data by PD-L1 expression and treatment indication. The revised title is:
“Treatment Strategies for First-Line PD-L1–Unselected Advanced NSCLC: A Comparative Review of Immunotherapy-Based Regimens by PD-L1 Expression and Clinical Indication”- The results are unclear, it would be much clearer to present them according to the studies indications and PD-L1 expression.
- Author response: We appreciate the reviewer’s constructive feedback. To improve clarity, we have restructured the Results section by organizing the findings according to treatment indications (e.g., ICI monotherapy, chemo-ICI combinations, dual-ICI regimens) and stratifying them by available PD-L1 expression subgroups (<1%, 1–49%, ≥50%). This revised format enables more meaningful comparisons across clinically relevant subpopulations. The updated structure and analyses can be found in the revised Results section (see lines 163–295). Corresponding Discussion was updated accordingly to reflect this organization.
- Please rewrite the conclusion according to the suggestions for the results.
- Author response: We thank the reviewer for the helpful comment. To improve clarity, we have reorganized the Results section by treatment indication (e.g., ICI monotherapy, chemo-ICI, dual-ICI) and stratified data by PD-L1 subgroups (<1%, 1–49%, ≥50%) to enable clearer comparisons. The revised content is available in the Conclusion (lines 421–440).
- Figures and tables are at the end of the manuscript and that is not according to the journal’s instructions. For all figures and table- it should be better described in the headings what the table/figure represents. In a table the column line of therapy is not necessary since all analyzed studies are in first line setting. In the column treatment the doses of the drugs are also unnecessary. In my opinion the table should be divided in two, one should contain the results for mono- immunotherapy and other for combination. Also it should be marked in the text the place of figures and tables.
- Author response: We thank the reviewer for these helpful recommendations. In accordance with the journal’s formatting guidelines, we have now moved all figures and tables to their appropriate positions within the main text and clearly indicated their placement. We have revised the table and figure captions to provide more informative descriptions of the content presented. The “line of therapy” column has been removed, as all studies are indeed first-line. Drug doses have also been removed from the “treatment” column for conciseness. We believe these adjustments improve clarity and presentation of the data.
- In some parts of the manuscripts the sentences are unclear, for example line 175- select monotherapy...
- Author response: We thank the reviewer for highlighting this issue. We have carefully revised the sentence at line 175 and other similar instances throughout the manuscript to improve clarity and readability. The sentence in question has been rephrased for better understanding and more precise expression. We have also conducted a thorough language review to ensure clarity and consistency across the manuscript.
Reviewer 2 Report
Comments and Suggestions for Authors
The manuscript provides a comprehensive and timely synthesis of current evidence on immunotherapy-based treatments for advanced NSCLC, particularly in PD-L1-unselected populations. The review is well-structured, methodologically rigorous, and addresses a clinically relevant topic. However, several areas require clarification or improvement to enhance the manuscript's impact and readability.
Major Comments:
- Could the title be revised to specify that the focus is on “first-line settings”?
- The second paragraph of the Introduction beginning with "Advancements in imaging technologies..." seems unrelated to the section on "PD-L1–Unselected Advanced NSCLC." Would you consider reorganizing or rewriting this portion for better alignment?
- In the “Results”section, please update all clinical trial data with the latest follow-up results. For example, include the 35-month follow-up of EMPOWER-Lung 1 (https://doi.org/10.1016/S1470-2045%2823%2900329-7).
- Still in the “Results”, please add the findings from the “phase 3 POSEIDON trial”regarding dual-ICI therapies (PMID: 39243945, https://doi.org/10.1016/j.jtho.2024.09.1381).
- In the “Discussion”, please consider adding content related to current treatment guidelines from “NCCN, ASCO, and ESMO”. Additionally, try to avoid repeating findings already detailed in the Results section.
- While the manuscript notes heterogeneity across trials (e.g., PD-L1 thresholds, chemotherapy regimens), it does not fully explore the implications. A subgroup analysis—such as by “histology”or “PD-L1 expression subgroups”—or a “meta-regression”could significantly strengthen the conclusions. For example, refer to this recent review (PMID: 39762577, https://doi.org/10.1038/s41571-024-00979-8) to discuss differences in ICI-based therapy efficacy across NSCLC subtypes.
- The discussion of biomarkers (e.g., PD-L1, TMB) is quite superficial. Please expand on “emerging biomarkers and technologies”(e.g., spatial transcriptomics, liquid biopsies) that may enhance patient selection beyond PD-L1. Consider highlighting ongoing trials that are exploring novel biomarkers and future directions.
- The “tables and figures”(especially Table 1 and Figures 2–4) are critical to the manuscript but are currently poorly formatted. For example, the structure of Table 1 makes it difficult to compare treatment regimens. Consider reorganizing the layout to improve clarity, perhaps by grouping treatments into categories such as “monotherapy”, “chemo-immunotherapy”, and “dual-ICI therapy”.
Author Response
-
Reviewer 2
- Could the title be revised to specify that the focus is on “first-line settings”?
- Author response: We thank the reviewer for the helpful suggestion. To clearly reflect the scope of the manuscript, we have revised the title to explicitly include the “first-line” setting. The updated title is:” Treatment Strategies for First-Line PD-L1–Unselected Advanced NSCLC: A Comparative Review of Immunotherapy-Based Regimens by PD-L1 Expression and Clinical Indication”
- The second paragraph of the Introduction beginning with "Advancements in imaging technologies..." seems unrelated to the section on "PD-L1–Unselected Advanced NSCLC." Would you consider reorganizing or rewriting this portion for better alignment?
- Author response: We thank the reviewer for this thoughtful observation. We agree that the paragraph beginning with “Advancements in imaging technologies...” appeared tangential in the context of PD-L1–unselected advanced NSCLC. We have revised this section to improve its relevance and ensure it aligns more directly with the focus of the manuscript (lines 48–54).
- In the “Results”section, please update all clinical trial data with the latest follow-up results. For example, include the 35-month follow-up of EMPOWER-Lung 1 (https://doi.org/10.1016/S1470-2045%2823%2900329-7).
- Author response: We thank the reviewer for this important suggestion. We have reviewed all included trials and updated the Results section with the most recent follow-up data available from published sources. Specifically, we have incorporated the 35-month follow-up results for EMPOWER-Lung 1 (Lancet Oncol. 2023;24:1314–1325) as suggested. Similar updates have been made for other trials where newer data were available. These revisions are reflected in the updated Results section
- Still in the “Results”, please add the findings from the “phase 3 POSEIDON trial”regarding dual-ICI therapies (PMID: 39243945, https://doi.org/10.1016/j.jtho.2024.09.1381).
- Author response: We thank the reviewer for the suggestion. We have added the updated findings from the phase 3 POSEIDON trial (PMID: 39243945) to the Results section, including efficacy and safety data for the durvalumab plus tremelimumab plus chemotherapy arm.
- In the “Discussion”, please consider adding content related to current treatment guidelines from “NCCN, ASCO, and ESMO”. Additionally, try to avoid repeating findings already detailed in the Results section.
- Author response: We appreciate the reviewer’s helpful feedback. In response, we have revised the Discussion with brief reccomandation from NCCN, ASCO, and ESMO to contextualize our findings (see lines 359–365).
- While the manuscript notes heterogeneity across trials (e.g., PD-L1 thresholds, chemotherapy regimens), it does not fully explore the implications. A subgroup analysis—such as by “histology”or “PD-L1 expression subgroups”—or a “meta-regression”could significantly strengthen the conclusions. For example, refer to this recent review (PMID: 39762577, https://doi.org/10.1038/s41571-024-00979-8) to discuss differences in ICI-based therapy efficacy across NSCLC subtypes.
- Author response: We thank the reviewer for this insightful suggestion. We have now expanded our analysis by presenting subgroup findings stratified by PD-L1 expression levels and histology where data were available. These additions are detailed in the Results section (see lines 163–295). We also incorporated discussion of relevant findings from the cited review (PMID: 39762577) to highlight how NSCLC subtypes may influence ICI efficacy.
- The discussion of biomarkers (e.g., PD-L1, TMB) is quite superficial. Please expand on “emerging biomarkers and technologies”(e.g., spatial transcriptomics, liquid biopsies) that may enhance patient selection beyond PD-L1. Consider highlighting ongoing trials that are exploring novel biomarkers and future directions.
- Author response: We appreciate the reviewer’s valuable suggestion. In response, we have expanded the biomarker discussion to include emerging approaches such as spatial transcriptomics, circulating tumor DNA, and multiplex immunohistochemistry, which hold promise for refining patient selection beyond PD-L1. We have also referenced ongoing clinical trials investigating novel biomarker strategies. These additions are included in the revised Discussion section (see lines 395–409).
- The “tables and figures”(especially Table 1 and Figures 2–4) are critical to the manuscript but are currently poorly formatted. For example, the structure of Table 1 makes it difficult to compare treatment regimens. Consider reorganizing the layout to improve clarity, perhaps by grouping treatments into categories such as “monotherapy”, “chemo-immunotherapy”, and “dual-ICI therapy”.
- Author response: We thank the reviewer for this thoughtful suggestion. While we agree that reorganizing Table 1 and Figures 2–4 by treatment categories (e.g., monotherapy, chemo-immunotherapy, dual-ICI therapy) could improve clarity, we chose to retain the current unified format to maintain a consistent trial-by-trial comparison across all regimens. This structure allows readers to assess study characteristics and outcomes in context without separating interrelated data. However, we have revised the table and figure captions to better guide interpretation and have improved formatting for clarity.
Round 2
Reviewer 1 Report
Comments and Suggestions for Authors
I would like to the authors for making the suggested corrections which significantly improved the article.
I have one small suggestion, please add abbreviations at the bottom of the tables.
Kind regards
Author Response
Comment: I would like to the authors for making the suggested corrections which significantly improved the article.
Response: Thank you very much for your kind words and constructive feedback. We truly appreciate your careful review and suggestions, which have helped us significantly improve the quality of our article.
Comment: I have one small suggestion, please add abbreviations at the bottom of the tables.
Response: Thank you for your valuable suggestion. We have added the abbreviations at the bottom of the tables as recommended.
Reviewer 2 Report
Comments and Suggestions for Authors
I appreciate the authors’ efforts in revising the manuscript; however, I do not believe the revisions have adequately addressed my previous concerns. Several key recent clinical trial updates remain missing, such as the latest follow-up results from CheckMate 9LA and POSEIDON, which significantly limits the article’s relevance and prevents it from being up-to-date.
Moreover, the conclusions presented in the review are already widely adopted in current clinical practice, and the manuscript lacks sufficient novelty. The authors have not provided an in-depth analysis of the existing limitations in the use of immunotherapy for NSCLC in clinical settings, nor have they proposed any constructive perspectives or forward-looking insights.
Given these limitations in terms of both content completeness and originality, I do not consider the manuscript suitable for publication.
Author Response
- Comment: I appreciate the authors’ efforts in revising the manuscript; however, I do not believe the revisions have adequately addressed my previous concerns. Several key recent clinical trial updates remain missing, such as the latest follow-up results from CheckMate 9LA and POSEIDON, which significantly limits the article’s relevance and prevents it from being up-to-date.
Response: Thank you for your valuable feedback and for highlighting the importance of including the latest clinical trial updates. We apologize for the oversight and have now incorporated the most recent follow-up results from CheckMate 9LA and POSEIDON in the revised manuscript. We believe these additions significantly enhance the relevance and currency of our article.
- Comment: Moreover, the conclusions presented in the review are already widely adopted in current clinical practice, and the manuscript lacks sufficient novelty. The authors have not provided an in-depth analysis of the existing limitations in the use of immunotherapy for NSCLC in clinical settings, nor have they proposed any constructive perspectives or forward-looking insights.
Response: Thank you for your insightful comments regarding the manuscript’s novelty and depth. We appreciate your perspective and have revised the conclusions to better highlight emerging challenges and limitations in the clinical use of immunotherapy for NSCLC. Additionally, we have included a new section offering forward-looking insights and potential future directions to provide a more comprehensive and constructive analysis.
- Comment: Given these limitations in terms of both content completeness and originality, I do not consider the manuscript suitable for publication.
Response: Thank you for your constructive feedback. We understand your concerns regarding content completeness and originality. We sincerely appreciate the opportunity to revise our manuscript further and address these limitations. We are committed to enhancing the quality and novelty of the work and hope that our revisions will better meet the standards for publication.